# Effects of biofertilizers on the growth, leaf physiological indices and chlorophyll fluorescence response of spinach seedlings

**Beibei Zhang[1,2]\*, Hui Zhang[1], Di Lu[3], Liping Cheng[3], Jiajia Li[3]**

**1** Shaanxi Key Laboratory of Disaster Monitoring and Mechanism Simulating, College of Geography and Environment, Baoji University of Arts and Sciences, Baoji, Shaanxi, China, **2** School of Surveying and Land Information Engineering, Henan Polytechnic University, Jiaozuo, Henan, China, **3** College of Tourism and Management, Pingdingshan University, Pingdingshan, Henan, China

\* zbb83101@126.com

**Data Availability Statement:** All relevant data are within the paper and its Supporting Information files.

## Abstract

Chemcial fertilizer as the main strategy for improving the vegetable yields was excessively applied in recent years which led to progressively serious soil problems such as the soil acidification. According the situation, five different biofertilizer treatments [no fertilizer (CK), inoculations of *Bacillus subtilis* (*Bs*, T1), combination of *Bs* and *Bacillus mucilaginosus* (*Bs* +*Bm*, T2), *Bs* and *Bacillus amyloliquefaciens* (*Bs+Ba*, T3), and *Bm+Ba* (T4)] were conducted to investigate the effect of the growth, leaf physiological indices, and chlorophyll fluorescence of spinach seedlings in the growth chamber. The growth and physiological indices of the spinach seedlings attained a maximum under the T2 treatments. Under the T2 treatment, the ABS/RC (Absorption flux per RC), $TR_0/RC$ (Trapping flux per RC), and $ET_0/RC$ (Electron transport flux per RC) was significantly increased, while the $DI_0/RC$ [Dissipated energy flux per RC (at t = 0)] was decreased. The OJIP curve was improved under of the inoculations of fertilizers, and the increasing range was the largest under the T2 treatment. The leaf light response curve (LC) was also significantly increased under the T2 treatment. The plant growth characteristics [leaf length (LL), leaf weight (LW), plant height (PH)] were positively correlated with the J-I-P test chlorophyll fluorescence parameters [$PI_{ABS}$ (Performance index for energy conservation from exciton to the reduction of intersystem electron acceptors), $\varphi P_0$ (Maximum quantum yield of primary photochemistry), $\varphi E_0$ (Quantum yield of electron transport), $\psi_0$ (The probability that a trapped exciton moved an electron in electron transport chain further than $QA^-$), $TR_0/RC$, and $ET_0/RC$] while negatively correlated with $\varphi D_0$ (Quantum yield of energy dissipation) and $DI_0/RC$. The leaf physiological characteristics [SP (soluble protein concentrations), SC (soluble carbohydrate concentrations), Chl a (chlorophyll a), Chl b (chlorophyll b), Chl a+b, Chl a/b, and WP (water potential)] were positively correlated with the J-I-P test chlorophyll fluorescence parameters ($PI_{ABS}$, $\varphi P_0$, $\varphi E_0$, $\psi_0$, ABS/RC, $TR_0/RC$, and $ET_0/RC$) while negatively correlated with $\varphi D_0$ and $DI_0/RC$. These results indicated that the combination of *Bs+Bm* inoculations promoted the growth of the spinach and improved the adaptability of the vegetable to acid soil while *Ba* inoculation didn't have any effects to plants.

**Funding:** This work was supported by the National Natural Science Foundation of China (NSFC, No. 41601016, 41771036), Special Support Program for High-Level Personnel Recruitment (Youth Top-Talent) of Shaanxi province, National Natural Science Foundation of Shaanxi province (No. 2015JM4136), Doctoral research start-up fee of Pingdingshan University (PXY-BSQD-2022039), Key projects of Shaanxi Provincial Department of Education(20JS011). The funders had no role in study design, data collection and analysis, decision to publish, or preparation of the manuscript.

**Competing interests:** The authors have declared that no competing interests exist.

## Introduction

Spinach (*Spinacia oleracea L.*) is a big-leaf variety, which belonged to the spinach genus Chenopodiaceae (annual herbaceous plant), as an important leafy green vegetable its leaves and shoots that contain large quantities of bioactive compounds, minerals, and vitamins like A, B, and C and nutrients [1, 2]. It can scavenge free radicals and has several medical and food applications [3]. The spinach plant has also several antibacterial compounds and folic acid which is useful for the treatment of anemia [4]. In 2020, the global spinach planting area is about 929, 000 hectares and the production is about 31.43 million tons [5, 6]. Among them, the global spinach production mainly came from China and Chinese spinach production was accounting for 91.51% of the global output [7]. So, the soil condition seriously affects the growth and yield of spinach. Recent years, fertilization as the main strategy to improve the vegetables' yield, the excessive application of chemical fertilizers has led to progressively serious soil problems such as the soil acidification, while the spinach is sensitive to it [8]. Therefore, biofertilizers have gradually replaced chemical fertilizers to improve soil quality to increase vegetable yield. Biofertilizers are also referred to as bacterial fertilizer, biological fertilizer, and microbial inoculant [9], which contains beneficial microorganisms that provide fertilization for crops through their activities.

The metabolic activities of microorganisms can improve soil fertility and crop quality, and also increase soil microorganism populations, reduce plant diseases, and enhance plant root activities [10, 11]. Among them, *Bacillus* is a type of growth-promoting bacteria that can produce phytase at the rhizosphere of plants. It has strong resistance to ultraviolet light, high salt, high acid, high heat, and radiation, with the capacity to inhibit bacteria, prevent disease, and increase production [12]. *Bacillus subtilis (Bs)* can improve the stress resistance of plants [13] and the availability of nitrogen and phosphorus in soil [14]. *Bacillus mucilaginosus (Bm)* can convert the unusable phosphorus in soil to available phosphorus for plants. Simultaneously, *Bm* can secrete substances that promote plant growth and development, such as growth hormones, gibberellin, and more [15, 16]. *Bacillus amyloliquefaciens (Ba)* has a certain antagonistic effect on bacterial diseases [17].

Photosynthesis as a process, through it plants can convert captured light energy into biochemical energy [18]. Photosynthetic performance was found to be a very informative indicator, because of its extreme sensitivity to environment [19].Chlorophyll fluorescence measurements as one of photosynthetic indicators have become a widely used method to study the functioning of the photosynthetic apparatus and are a powerful tool to study the plant's response to different environments [20]. Through the analyses of chlorophyll fluorescence parameters, a further elucidation of the light energy absorption, utilization, transfer, and dissipation of plant chloroplast Photosystem I (PSI) and Photosystem II (PSII) processes can be obtained [21]. The chlorophyll fluorescence technique has also been widely investigated for the detection of physiological changes in plants that are caused by bacterial and fungal infections [22]. Chu et al. [23] and Bing et al. [24] revealed that there was a significant correlation between chlorophyll fluorescence parameters and the degree of *Verticillium* wilt infection in cotton leaves. Tung et al. [25] quantitatively analyzed the degree of infected tobacco (infected, unmanifested infection, healthy) according to Fv/Fm images, which indicated that chlorophyll fluorescence imaging technologies could detect the degree of infected tobacco.

The purpose of this research was to evaluate the variables related to growth, physiological characters and chlorophyll florescence of spinach seedlings in response to different biofertilizers under acid soil condition. The results of this study would be used to give guidance on growth, yield and soil improvement for spinach cultivation in a controlled environment.

## Materials and methods

### Experimental materials

The spinach variety *Spinacia oleracea L.*was used as the experimental material. The *Bacillus subtilis* (*Bs*), *Bacillus mucilaginosus* (*Bm*), and *Bacillus amyloliquefaciens* (*Ba*) selected the original strains produced by Lvlong Biotecnology Co., Ltd., which contained >200 million live bacteria per gram. The fertilizer components were the *Bacillus* and its metabolites, and the rate of miscellaneous bacteria <3%.

No specific permissions were required for research locations and the studies did not involve endangered or protected species. All the authors consent for publication.

**Growth conditions and experiment design.** The soil came from the Qinxi demonstration garden in Taibai County whose nutrients were comprised of 33.35 $g\cdot kg^{-1}$ organic matter, 2.79 $g\cdot kg^{-1}$ total nitrogen, 1.51 $g\cdot kg^{-1}$ total phosphorus, and 10.75 $mg\cdot kg^{-1}$ available phosphorus, and a pH of 5.2. The experiments were carried out in some growth chambers (FytoScope FS 130-WIR, Brno, Czech) under a 12 h day/night photoperiod at temperatures of 25/15˚C, respectively, a photosynthetically active radiation (PAR) of 300 $\mu mol/(m^2 \cdot s)$, and humidity of 60%. According to the experiment design, the growth chambers (FytoScope FS 130-WIR) could monitor, control and adjust the temperature, PAR and humidity automatically. Five different biofertilizer treatments [no fertilizer (CK), inoculations of *Bacillus subtilis* (*Bs*, T1), combination of *Bs* and *Bacillus mucilaginosus* (*Bs*+*Bm*, T2), *Bs* and *Bacillus amyloliquefaciens* (*Bs* +*Ba*, T3), and *Bm*+*Ba* (T4)] were conducted at March 10[th], 2021. The experimental design was a factorial scheme with 4 replicates per treatment.

The spinach seeds were soaked in deionized water for 12 h and placed in a 4˚C refrigerator for 24 h to promote germination. The germinated seeds were placed on a seedling tray, covered with nutritious soil, and germinated in the dark in a 25˚C incubator. Following seven days of treatment, the seedlings were transferred to pot (Ø25 cm x 20 cm high) filled with soil 18 cm deep, and 10 seedlings were planted in each pot. For this experiment, 90 $kg\cdot km^{-2}$ of phosphate fertilizer (potassium dihydrogen phosphate) was used as base fertilizer with biofertilizer 0.5 $kg\cdot km^{-2}$.

**Measurement of growth indices.** For this experiment, all of the characteristics of the spinach leaves were measured at 7[th] day after treatments. The plant height (PH), leaf length (LL) and leaf weight (LW) were measured with a scale.

**Measurement of leaf physiological indices.** After 7 to 10 days treatment, the leaf soluble protein concentration (SP) was determined by the Coomassie brilliant blue (CBB) method at 595 nm using a DR6000 spectrophotometer [26], whereas the soluble carbohydrate concentration (SC) was determined using an anthrone method at 620 nm [27]. The leaf Chlorophyll a (Chl a) and Chlorophyll b (Chl b) concentrations were determined by Litchtenthaler [28] at 645 and 663 nm, whereas the leaf malondialdehyde (MDA) was determined by the thiobarbituric acid method [29]. The leaf water potential (WP) (ψ) was measured by dew point potentiometer (WP4, Decagon Devices, Pullman, USA).

**Chlorophyll fluorescence measurements.** Prior to the fluorometer measurements, the leaves were dark-adapted for 20 min and then measured using a FluorPen FP 100Max hand fluorescence meter (Photon Systems Instruments, Brno, Czech), and calculated according to the JIP-test algorithm proposed by Strasser et al. [30] (Table 1).

To compare the OJIP curve with the normalized Chlorophyll fluorescence (CF) transient curve between OJ and OK, the following Formulas (1), (2), (3), (4), (5) and (6) were used.

$$V_t = (F_t - F_0)/(F_m - F_0) \tag{1}$$

**Table 1. Parameters of chlorophyll fluorescence.**

| Parameters and Formula | Meaning |
| --- | --- |
| $F_0$ | minimum fluorescence |
| $F_m$ | maximum fluorescence |
| $F_J$ | $F_t$ at Time 2 ms |
| $F_I$ | $F_t$ at Time 30 ms |
| $F_t$ | The relative fluorescence intensity at Time t |
| $F_v = F_m - F_0$ | variable fluorescence |
| $nF_v = F_v/F_0$ | variable to initial fluorescence ratio |
| $V_I = (F_J - F_0)/(F_m - F_0)$ | Relative variable fluorescence at the I-step |
| $V_J = (F_J - F_0)/(F_m - F_0)$ | Relative variable fluorescence at the J-step |
| $M_0$ | Approximated initial slope (in $ms^{-1}$) of the fluorescence transient V = f(t) |
| $PI_{ABS}$ | Performance index (potential) for energy conservation from exciton to the reduction of intersystem electron acceptors |
| ABS/RC | Absorption flux (of antenna Chls) per RC |
| $TR_0/RC = M_0(1/V_J)$ | Trapping flux (leading to QA reduction) per RC |
| $ET_0/RC = M_0(1/V_J)\Psi o$ | Electron transport flux (further than $QA^-$) per RC |
| $DI_0/RC = (ABS/RC - TR_0/RC)$ | Dissipated energy flux per RC (at t = 0) |
| $\varphi P_0 = 1 - F_0/F_m$ | Maximum quantum yield of primary photochemistry (at t = 0) |
| $\varphi E_0 = (1 - F_0/F_m)(1 - V_J)$ | Quantum yield of electron transport (at t = 0) |
| $\Psi_0 = ET_0/Tr_0 = 1 - V_J$ | The probability that a trapped exciton moved an electron in electron transport chain further than $QA^-$ (t = 0) |
| $\varphi D_0 = F_0/F_m$ | Quantum yield (at t = 0) of energy dissipation |

$$\Delta V_t = V_t^{TR} - V_t^{CK} \tag{2}$$

$$W_{OJ} = (F_t - F_0)/(F_j - F_0) \tag{3}$$

$$\Delta W_{OJ} = W_{OJ}^{TR} - W_{OJ}^{CK} \tag{4}$$

$$W_{OK} = (F_t - F_0)/(F_K - F_0) \tag{5}$$

$$\Delta W_{OK} = W_{OK}^{TR} - W_{OK}^{CK} \tag{6}$$

The leaf light-response curve (LC) measurement, based on pulse modulated fluorometry (PAM) was designed using seven photosynthetic photon flux densities (PPFD) (10, 20, 50, 100, 300, 500, and 1000 μmol/(m²·s)) to acquire chlorophyll fluorescence parameter changing curves (Ft, QY) relating the rate of photosynthesis.

**Statistical analysis of data.** All collected data were subjected to one-way ANOVA analysis using SPSS (SPSS software version 22.0, Chicago, Illinois, USA). Differences between means were compared by Bonferroni test at $P < 0.05$. The correlations between parameters were determined using Pearson's simple correlation test function in SPSS.

## Results

### Effects of biofertilizers on growth and physiological indices of spinach seedlings

There were significant differences in plant height (PH), leaf length (LL), leaf weight (LW) (F = 54.37, 13.30, and 46.03, respectively; $P < 0.01$), where the original data of growth indices

**Table 2. Differences of the growth characteristics under 5 different biofertilizer treatments of spinach.**

| Treatments | Growth characters | | |
|---|---|---|---|
| | Plant height (PH, cm) | Leaf length (LL, cm) | Leaf weight (LW, g/10 plants) |
| None, CK | 3.53±0.15 b | 2.62±0.26 b | 0.95±0.01 c |
| *Bs*, T1 | 5.23±0.09 a | 4.08±0.05 a | 2.08±0.02 a |
| *Bs+Bm*, T2 | 5.27±0.09 a | 4.15±0.09 a | 2.69±0.02 a |
| *Bm+Ba*, T3 | 5.05±0.03 a | 3.73±0.18 a | 1.47±0.02 b |
| *Bs+Ba*, T4 | 5.07±0.03 a | 3.89±0.19 a | 1.74±0.01 b |
| F value | 54.37** | 13.30** | 46.03** |

Values presented in each column of table have a mean ±standard deviation. The last portion of table refers to the F value

\* P<0.05

\*\* P<0.01.

was maximum under the T2 treatment (Table 2). There were significant differences among the treatments in the leaf physiological indices (Table 3). Biofertilzer adding had impact on the leaf physiological responses especially under T2 treatment. Compared with the F value, the difference in soluble sugar (SC) was the highest (F = 118.35; P<0.01).

## Effect of biofertilizers on chlorophyll fluorescence parameters of spinach seedlings

According to the values of the basic fluorescence parameters (Table 4), the $F_0$ value of the T2 treatment was the highest, which was 41.29% higher than that of CK. Further, the $F_m$ value of the T2 treatment was 17751 higher than that of CK, and there was a significant difference in $F_V$ value under the five treatments. The relative variable fluorescence value ($V_I$) at 30 ms was highest under the T2 treatment. In terms of the $F_v/F_0$ value, the T2 treatment remained the largest.

There were significant differences between the $M_0$ and $PI_{ABS}$ values under the five different treatments (F = 7.66 and 4.09, respectively; P<0.05) (Table 5). There was a significant difference between the $TR_0/RC$ and $DI_0/RC$ under T2 treatment (Table 6). However, there was no

**Table 3. Differences of leaf physiological indexes under 5 different biofertilizer treatments of spinach.**

| Treatments | Leaf physiological indexes | | | | | | | |
|---|---|---|---|---|---|---|---|---|
| | SP | SC | MDA | Chl a | Chl b | Chl a+b | Chl a/b | WP($\Psi$, Mpa) |
| | (mg/g FW) | (mg/g FW) | (mg/gFW) | (mg/g FW) | (mg/g FW) | (mg/g FW) | (mg/g FW) | |
| None, CK | 1.83±0.26c | 0.73±0.05d | 5.49±0.51a | 1.16±0.03e | 0.36±0.03c | 1.52±0.03e | 3.25±0.30b | -6.57±0.25d |
| *Bs*, T1 | 3.31±0.20b | 2.26±0.06b | 1.80±0.13b | 2.73±0.22b | 0.58±0.06ab | 3.30±0.15b | 4.91±0.83a | -4.23±0.12b |
| *Bs+Bm*, T2 | 4.14±0.04a | 3.59±0.13a | 1.25±0.14b | 3.23±0.04a | 0.60±0.03a | 3.83±0.07a | 5.36±0.25a | -2.79±0.22a |
| *Bm+Ba*, T3 | 2.37±0.08c | 1.59±0.11b | 4.82±0.31a | 1.70±0.07d | 0.40±0.01c | 2.11±0.07d | 4.21±0.09ab | -5.53±0.26c |
| *Bs+Ba*, T4 | 3.28±0.22b | 2.23±0.11c | 2.09±0.15b | 2.29±0.11c | 0.47±0.02bc | 2.75±0.09c | 4.91±0.43a | -4.79±0.12b |
| F value | 24.48** | 118.35** | 44.95** | 50.63** | 8.80** | 118.34** | 203.88** | 48.72** |

Values presented in each column of table have a mean ±standard deviation. The last portion of table refers to the F value

\* P<0.05

\*\* P<0.01

SP: soluble protein concentrations, SC: soluble carbohydrate concentrations, MDA: malonaldehyde concentrations, Chl: chlorophyll, WP: water potential, FW: fresh weight.

**Table 4. Differences of leaf chlorophyll fluorescence intensity under 5 different biofertilizer treatments of spinach.**

| Treatments | Leaf chlrophyll fluorescence intensity (relative unit) | | | | | |
|---|---|---|---|---|---|---|
| | $F_0$ | $F_m$ | $F_v$ | $V_J$ | $V_I$ | $F_v/F_0(nF_v)$ |
| None, CK | 7087±188b | 28967±1440c | 21880±1628c | 0.46±0.06a | 0.76±0.01bc | 4.07±0.03 a |
| *Bs*, T1 | 9233±568ab | 43142±1342ab | 33909±1286ab | 0.54±0.03a | 0.82±0.01ab | 3.67±0.11a |
| *Bs+Bm*, T2 | 10013±10a | 46718±1126a | 36705±1136a | 0.47±0.00a | 0.85±0.02a | 4.08±0.11a |
| *Bm+Ba*, T3 | 9071±108ab | 31861±131bc | 22790±23bc | 0.47±0.01a | 0.76±0.00c | 2.51±0.41b |
| *Bs+Ba*, T4 | 9786±38ab | 41549±1544ab | 31763±1506abc | 0.46±0.00a | 0.77±0.01bc | 3.25±0.24a |
| F value | 112.57** | 31.13** | 24.01** | 2.48 | 3.85* | 8.3** |

The values presented in each column of table have a mean ±standard deviation. The last part of table refers to the F value

* $P < 0.05$

** $P < 0.01$.

significant difference in the ABS/RC and $ET_0/RC$ between the treatments. The quantum yield ($\varphi P_0$) and efficiency ($\varphi E_0$) values were highest under the T2 treatment, and with the exception of $\psi_0$. On the contrary, the value of $\varphi D_0$ was lowest under the T2 treatment.

## Effect of biofertilizers on transient analysis of prompt fluorescence OJIP of spinach seedlings

The transient curve of OJIP was shown in Fig 1 and the OJIP curves of the five treatments were similar to those reported by Strasser et al. [31]. In Fig 1(A), compared with CK, the fluorescence trend of other treatments gradually increased. Among them, during the O-J band, the fluorescence value of the T1 treatment was the largest, and From I-P band, the fluorescence value of the T2 treatment was the largest, which signified that it had a greater effect on the photosynthetic chemical rate of the leaves. As shown in Fig 1(B), the difference of relative variable fluorescence ($V_t$) at point J (2 ms) was the largest from the statically analysis (F = 2354.87, P<0.01). This was better illustrated in Fig 1(C) $\Delta V_t$. Over time the difference between the treatments increased initially, then decreased, and the difference of the initial slope during the O-J stage curve was maximized. It can be seen in Fig 1D and 1E, while there was an obvious positive K band under the T1 and T2 treatments, and a negative growth trend under the T3 and T4 treatments. The transient double normalization of OJIP between $F_0$ and $F_K$ and the difference of double normalization between the treatments and the control group were regarded as the L band. As shown in Fig 1F and 1G, the $W_{OK}$ value of the T2 treatment was higher, and there was a significant higher positive L band.

**Table 5. Differences of J-I-P test parameters under 5 different biofertilizer treatments of spinach leaf.**

| Treatments | J-I-P test parameters (relative unit) | | | | | |
|---|---|---|---|---|---|---|
| | $M_0$ | $PI_{ABS}$ | ABS/RC | $TR_0/RC$ | $ET_0/RC$ | $DI_0/RC$ |
| None, CK | 0.91±0.03ab | 0.38±0.29c | 2.20±0.08a | 1.37±0.06ab | 0.46±0.04a | 0.82±0.06a |
| *Bs*, T1 | 1.08±0.07a | 1.24±0.21ab | 2.57±0.03a | 2.02±0.01ab | 0.94±0.03a | 0.55±0.02c |
| *Bs+Bm*, T2 | 1.00±0.02a | 1.70±0.23a | 2.59±0.03a | 2.08±0.04a | 1.07±0.01a | 0.51±0.01b |
| *Bm+Ba*, T3 | 0.67±0.01ab | 1.41±0.01bc | 2.01±0.00a | 1.43±0.02b | 0.76±0.01a | 0.57±0.01ab |
| *Bs+Ba*, T4 | 0.83±001b | 1.61±0.03bc | 2.35±0.08a | 1.80±0.01ab | 0.96±0.01a | 0.55±0.00bc |
| F value | 7.66** | 4.09* | 2.88 | 3.97* | 2.04 | 8.16** |

The values presented in each column of table have a mean ±standard deviation. The last part of table refers to the F value

* $P < 0.05$

** $P < 0.01$.

**Table 6. Differences of leaf chlorophyll fluorescence yield and efficiency under 5 different biofertilizer treatments of spinach.**

| Treatments | Yield and efficiency | | | |
|---|---|---|---|---|
| | $\varphi P_0 (F_v/F_m)$ | $\varphi E_0$ | $\Psi_0$ | $\varphi D_0$ |
| None, CK | 0.62±0.01c | 0.22±0.01a | 0.34±0.01a | 0.37±0.01ab |
| *Bs*, T1 | 0.78±0.01ab | 0.37±0.01a | 0.47±0.01a | 0.21±0.00b |
| *Bs+Bm*, T2 | 0.80±0.00a | 0.41±0.06a | 0.52±0.05a | 0.19±0.00cc |
| *Bm+Ba*, T3 | 0.72±0.0bc | 0.38±0.03a | 0.53±0.02a | 0.29±0.03a |
| *Bs+Ba*, T4 | 0.76±0.01ab | 0.41±0.00a | 0.54±0.00a | 0.24±0.01bc |
| F value | 6.03** | 2.48 | 2.71 | 6.02** |

The values presented in each column of table have a mean ±standard deviation. The last part of table refers to the F value

* P<0.05

** P<0.01.

## Effect of biofertilizers on the difference of the light response curve (LC) of the spinach seedlings

In Fig 2, compared with the five treatments, the quantum yield (QY) of leaves under the T2 treatment attained a maximum at about 20 μmol/(m$^2$·s) PPFD, which indicated that the light compensation point (LCP) was around 20 μmol/(m$^2$·s) PPFD. The maximum value of the other treatments was before 10 μmol/(m$^2$·s) PPFD, indicating that the LCP was earlier than 10 μmol/(m$^2$·s) PPFD.

## Correlations between growth, leaf physiological indices and chlorophyll fluorescence

Through correlation analyses, the indices of growth, physiology, and chlorophyll fluorescence of leaves were quantified. Plant growth indices (LL, LW, PH) were positively correlated with chlorophyll fluorescence parameters (PI$_{ABS}$, $\varphi P_0$, $\varphi E_0$, $\psi_0$, TR$_0$/RC, ET$_0$/RC), while which was negatively correlated with $\varphi D_0$ and DI$_0$/RC. The physiological indices of the leaves (SP, SC, Chl a, Chl b, Chl a+b, Chl a/b, and WP) were positively correlated with PIABS, $\varphi P_0$, $\varphi E_0$, $\psi_0$, ABS/RC, TR$_0$/RC and ET$_0$/RC, and negatively correlated with $\varphi D_0$ and DI$_0$/RC. The leaf MDA was significantly positively correlated with $\varphi D_0$ and DI$_0$/RC, and positively correlated with other chlorophyll fluorescence parameters (Table 7).

## Discussion

### Physiological indices of growth and leaves

In terms of plant phenology and biological characteristics, plant growth, leaf chlorophyll concentration, and other physiological indices were critical influencing factors [32]. The results of Yan et al. [33] revealed that the application of biofertilizer could effectively improve the plant height of pakchoi. In this study, the inoculations of biofertilizer effectively improved the plant height, leaf length, and leaf weight of spinach. The spinach plants growth status under the inoculations of *Bs* or *Bm* was improved and it was similar to the results of Samia et al. [34].

The leaf soluble protein (SP), soluble carbohydrate (SC), and water potential (WP) were closely related to plant metabolism. Chlorophyll (Chl) was an important substance in plant photosynthesis, and malondialdehyde (MDA) can reflect the degree of membrane lipid peroxidation [35]. In this research, When *Bs* and *Bm* (T2) were inoculated together, the leaf physiological indices of spinach seedling (SP, SC, Chl a, Chl b, Chl a+b, Chl a/b, and WP) were

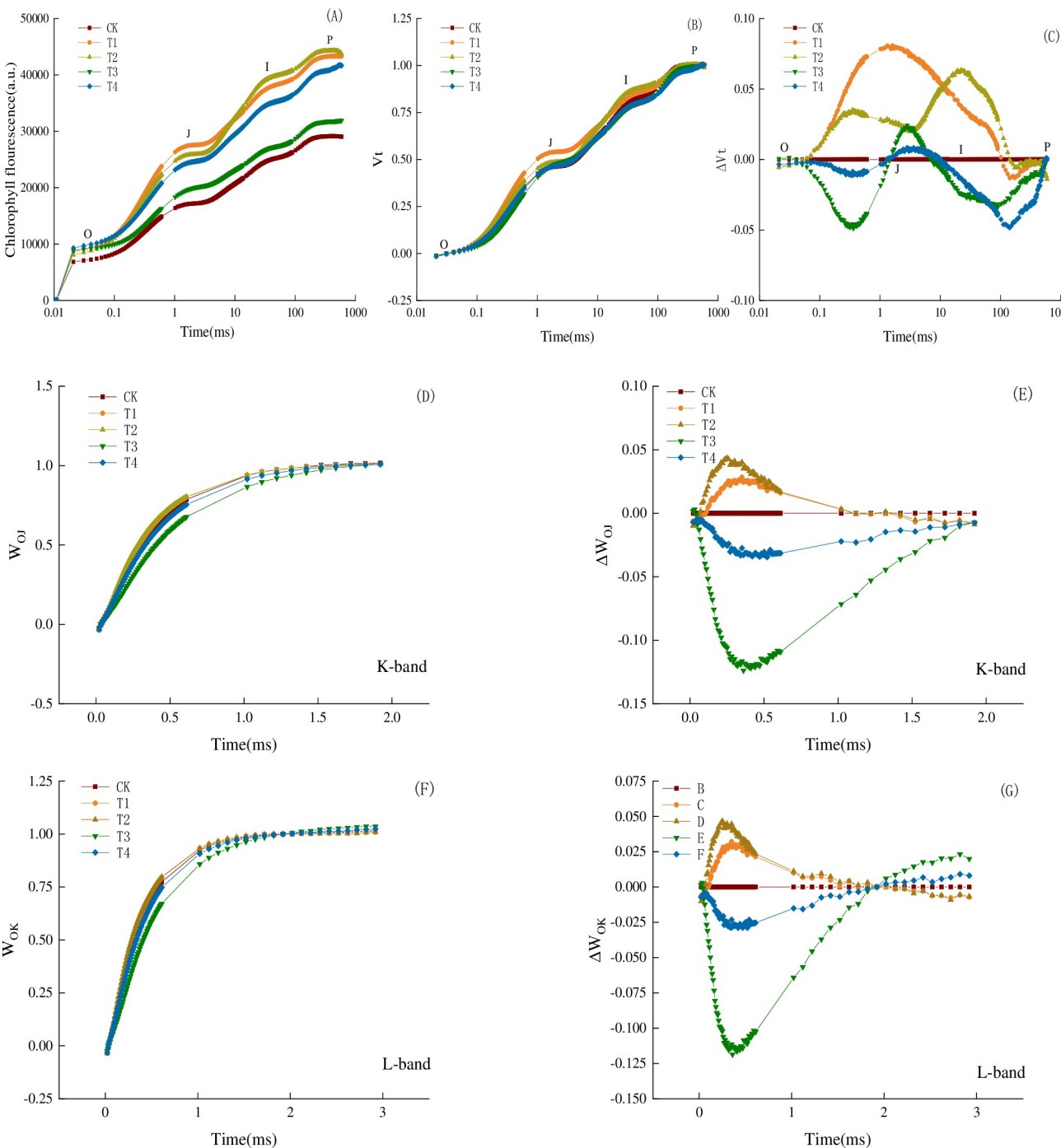

**Fig 1. Shape change of leaf chlorophyll fluorescence transient curve under 5 different biofertilizer treatments.** Each curve represents the average of three independent measurements. (A) Relative chlorophyll fluorescence intensity transient curves; (B) Relative variable fluorescence intensity transient curves ($V_t$); (C) Normalized the relative variable fluorescence intensity transient curves ($\Delta V_t$); (D) Relative variable fluorescence intensity transient curves of the O-J segments ($W_{OJ}$); (E) Normalized the relative variable fluorescence intensity transient curves of the O-J segments ($\Delta W_{OJ}$); (F) Relative variable fluorescence intensity transient curves of the O-K segments ($W_{OK}$); (G) Normalized the relative variable fluorescence intensity transient curves of the O-K segments ($\Delta W_{OK}$). No fertilizer (CK), inoculations of *Bacillus subtilis* (*Bs*, T1), combination of *Bs* and *Bacillus mucilaginosus* (*Bs+Bm*, T2), *Bs* and *Bacillus amyloliquefaciens* (*Bs+Ba*, T3), and *Bm+Ba* (T4).

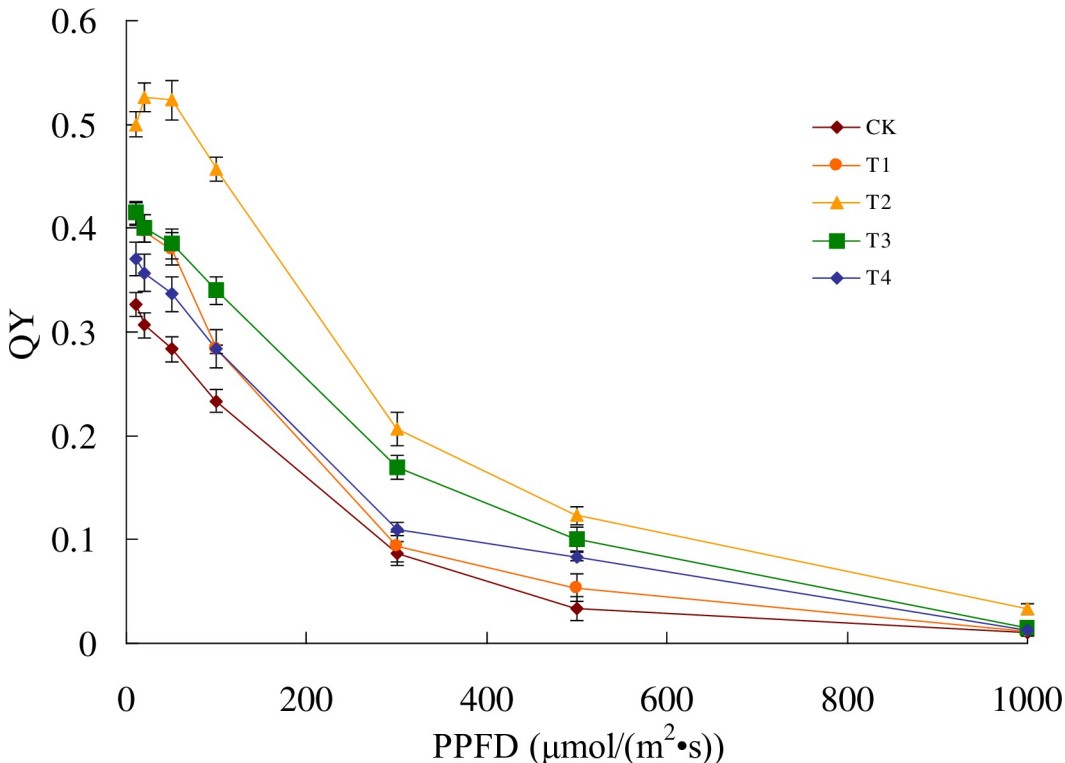

**Fig 2. Light curve (LC) of quantum yield (QY) changes under 5 biofertilizer treatments.** Data are the means of three experiments. No fertilizer (CK), inoculations of *Bacillus subtilis* (*Bs*, T1), combination of *Bs* and *Bacillus mucilaginosus* (*Bs*+*Bm*, T2), *Bs* and *Bacillus amyloliquefaciens* (*Bs*+*Ba*, T3), and *Bm*+*Ba* (T4).

significantly enhanced. Wang et al. [36] found that the application of biofertilizer could effectively increase the chlorophyll concentration of winter wheat at the different growth stages.

For this study, the leaf chlorophyll concentration treated with the combination of *Bs*+*Bm* treatment (T2) was the highest, whereas when *Bs* or *Bm* were paired with *Ba* there was less chlorophyll. It might be possibly explained that it could convert ineffective nitrogen or

**Table 7. Correlations between the leaf physiological characteristics and chlorophyll fluorescence parameters.**

| Index | LW | LL | PH | SP | SC | Chl a | Chl b | Chl a+b | Chl a/b | MDA | WP |
|-------|------|------|------|------|------|--------|--------|---------|---------|---------|--------|
| PIABS | 0.75** | 0.84** | 0.92** | 0.76** | 0.81** | 0.73** | 0.55* | 0.73** | 0.68** | -0.71** | 0.75** |
| $\varphi P_0$ | 0.90** | 0.89** | 0.91** | 0.89** | 0.90** | 0.93** | 0.80** | 0.93** | 0.74** | -0.91** | 0.90** |
| $\varphi E_0$ | 0.58* | 0.78** | 0.89** | 0.60* | 0.65** | 0.56* | 0.38 | 0.55* | 0.58* | -0.55* | 0.58* |
| $\Psi_0$ | 0.74** | 0.86** | 0.95** | 0.75** | 0.79** | 0.74** | 0.56* | 0.74** | 0.68** | -0.72** | 0.74** |
| $\varphi D_0$ | -0.90** | -0.89** | -0.91** | -0.89** | -0.90** | -0.93** | -0.80** | -0.93** | -0.74** | 0.91** | -0.90** |
| ABS/RC | 0.77** | 0.51 | 0.40 | 0.78** | 0.73** | 0.83** | 0.80** | 0.84** | 0.54* | -0.85** | 0.78** |
| $TR_0$/RC | 0.90** | 0.74** | 0.69** | 0.90** | 0.88** | 0.94** | 0.86** | 0.96** | 0.68** | -0.95** | 0.90** |
| $ET_0$/RC | 0.89** | 0.88** | 0.90** | 0.91** | 0.92** | 0.91** | 0.76** | 0.92** | 0.75** | -0.91** | 0.90** |
| $DI_0$/RC | -0.80** | -0.90** | -0.97** | -0.78** | -0.82** | -0.80** | -0.65** | -0.81** | -0.70** | 0.76** | -0.79** |

* $P < 0.05$

** $P < 0.01$

LL: Leaf length, LW: Leaf weight, PH: Plant height, SP: Soluble protein concentrations, SC: Soluble carbohydrate concentrations, MDA: malonaldehyde concentrations, Chl: chlorophyll, WP: Water potential.

phosphorus into available nitrogen or phosphorus in the soil and improved the increase in N fixation and P solubilization of the plants after the application of *Bs* or *Bm*. It also maybe promoted the synthesis of ATP (Adenosine triphosphate) and NADPH (nicotinamide adenine dinucleotide phosphate) in leaves then promoting the plant growth. Likewise, from Awan's results, after inoculation it also could affect the regulation of phytohormone biosynthesis pathways; modulate ethylene levels in plants and the launch of host plants' systemic tolerance [37]. The inoculation of *Ba* didn't have more effects. The reason for the analysis may be that *Ba* mainly targets the repair effect of plant facing to bacterial and fungal infections, and the plants used in the experiment were not infected, so the promoting effect on plant growth is not significant. When plants were under stress, the MDA concentration was typically an important indicator of membrane lipid peroxidation, which reflected the harmful effects of stress on plant cells and tissues [38]. In this study, the concentration of MDA in leaves treated with biofertilizer was decreased which mean the plant was in a suitable environment.

## Leaf chlorophyll fluorescence

In this research, the values of $F_0$ and Fm were enhanced, which indicated that the *Bs* and *Bm* promoted an increase in the size of leaf PSII antenna and a decrease in the non-radiative dissipation of chlorophyll in PSII antenna, thus increasing the capacity of leaves to capture light energy [39]. We observed higher Fv and $F_v/F_0$ ($nF_v$) values under the combination of *Bs*+*Bm* treatment, which signified an increase in the efficiency of supplying electrons to PSII reaction centre (RCS) and the photosynthetic quantum conversion rate of PSII RCS. This translated to less energy being used for non-photochemical dissipation in the dark adaptation process [40].

In this experiment, the ABS/RC values of the leaves increased, indicating that after inculcating *Bs*+*Bm* the size of active RCS was increased, which led to a higher number of active PSII reaction centers and enhanced dark accumulation [20]. Under the action of the *Bs*+*Bm*, the $TR_0/RC$ value increased, which reflected the higher electron capture rate of RC, where more QA (the primary electron acceptor in Photosystem II) was converted to $QA^-$. This resulted in an increase in the electron transfer energy ($ET_0/RC$), thus reducing the energy dissipated by non-photochemical activities ($DI_0/RC$) [41, 42].

The $\varphi P_0$ (Fv/Fm) represented the maximum quantum yield of PS II, and the $\psi_0$ value reflected the electron transfer efficiency, from $QA^-$ to QB, whereas $\varphi E_0$ reflected the quantum yield of electron transport. The increase of $\varphi P_0$, $\psi_0$, and $\varphi E_0$ indicated that the combination of *Bs*+*Bm* promoted the redox reaction following QA, which resulted in an increase of the electron transfer rate between $QA^-$ and QB [43].

Oukarroum et al. [39] regarded the double normalization of the OJIP transient between the peak value of 0.05 to 2 ms, and the difference of double normalization between the treatment and the control group as the K-band. The change of the chlorophyll fluorescence curve was closely related to the physiological morphology of plants [44]. Changes in the O-J segment were related to an increased number of inactive reaction centers, or the energy transfer from the LHC II (Light-harvesting Complex) to PS II RCS [45]. The K and L bands reflected the connection between the S state of the PSII oxygen evolution complex (OEC) and PS II unit, as well as the energy connection between the PS II units [46, 47].

An increase in the J-I segment could reflect a decrease in the relative number of active PQ (plastoquinone) molecules that was reduced by each active RCS of PSII [47]. Further, changes in the I-P segment were closely related to the pool of electron receptors (ferredoxin and NADPH) at the end of the PSI, signifying the kinetic flow rate to the electron receptors at the end of the PSI [48]. In this experiment, the OJIP transient curves of the spinach seedlings were affected by biofertilizer (Fig 1).

Under the T2 treatment, the relative fluorescence intensity of chlorophyll in the O-J segment exhibited a larger value, indicating that the population of active PSII centers decreased, whereas the QA$^-$ accumulated massively. (Fig 1E and 1G) showed that there were obvious K bands and positive L bands under the T2 treatment, which indicated that the PSII had an inhibitory effect on the OEC. This resulted in a weakening of the connectivity between PSII and OEC and a decreased energy connection between the PSII units. This might due to the variable light and ventilation that was present at different locations in the incubator, which might have led to PSI receptor side damage and chlorophyll protein denaturation in some plant leaves [49]. In addition, the fluorescence of J-I and I-P segments remained large under the T2 treatment, which indicated that the relative number of PQ decreased, while there was an increase in the pool of electron receptors (ferredoxin and NADPH) at the end of PSI, which led to a higher kinetic flow rate to the electron receptors at the end of PSI. Combined with PI$_{ABS}$, the higher PI$_{ABS}$ values under combination of *Bs* and *Bm* treatment indicated increases in the density of active PSII centers, the efficiency of photoreactions, and efficiency of biochemical dark redox reactions, as well as the production and utilization of NADPH [50].

## Conclusions

In this study, the role of different biofertilizers was analyzed in spinach seedling. It was observed that the inoculations of biofertilizers *Bs* or *Bm* stabilized the plant cell membrane and increased the rate of photosynthesis which led toward the improvement of plant growth. Acid soil reduced the growth of plant and increased the amount of ROS production which disrupted the plant metabolism. The application of biofertilizers triggered osmotic adjustment, maintained the production of antioxidants, improved the leaf chlorophyll fluorescence and promoted the growth in plants. The T2 treatment significantly ($p < 0.05$) increased the growth, physiological index and leaf chlorophyll fluorescence of spinach. It could be concluded that combined inoculations of *Bs+Bm* was a better strategy for reducing the negative effects of acid soil on spinach. Due to limited experimental conditions, the experiment was only analyzed at 300 μmol/(m$^2$•s) light conditions without designing the multiple light conditions. In the next experiment, the impact of biofertilizer inoculation on spinach under multiple light conditions will be explored. Likewise, future research also should endeavor to analyze the utilization of mineral elements in the soil, prevent soil acidification caused by the excessive accumulation of mineral elements and spinach quality to control the fertilization concentrations of microorganisms, so as to provide an improved theoretical basis for spinach production and obtained the great economic benefits.

## Supporting information

**S1 Data.**
(XLSX)

## Acknowledgments

The authors are grateful to Jing Qi, Jiaxuan Wang, Jiayi Han and Xiaoying Hu for assistance with the data measuring.

## Author Contributions

**Conceptualization:** Beibei Zhang.

**Data curation:** Liping Cheng, Jiajia Li.

**Formal analysis:** Hui Zhang.

**Funding acquisition:** Beibei Zhang.

**Investigation:** Di Lu.

**Writing – original draft:** Beibei Zhang, Hui Zhang.

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
