## [Decision Letter · Decision Letter 0]

27 Jul 2023

PONE-D-23-13632Effects of biofertilizers on the growth, leaf physiological Indices and chlorophyll fluorescence response of spinach seedlingsPLOS ONE

Dear Dr. Zhang,

Thank you for submitting your manuscript to PLOS ONE. After careful consideration, we feel that it has merit but does not fully meet PLOS ONE’s publication criteria as it currently stands. Therefore, we invite you to submit a revised version of the manuscript that addresses the points raised during the review process.

We look forward to receiving your revised manuscript.

Kind regards,

Mayank Gururani

Academic Editor

PLOS ONE

Journal Requirements:

"NO. The funders had no role in study design, data collection and analysis, decision to publish, or preparation of the manuscript."

Reviewers' comments:

Reviewer's Responses to Questions

**Comments to the Author**

1. Is the manuscript technically sound, and do the data support the conclusions?

Reviewer #1: Yes

Reviewer #2: Partly

2. Has the statistical analysis been performed appropriately and rigorously? 

Reviewer #1: Yes

Reviewer #2: Yes

3. Have the authors made all data underlying the findings in their manuscript fully available?

Reviewer #1: Yes

Reviewer #2: Yes

4. Is the manuscript presented in an intelligible fashion and written in standard English?

Reviewer #1: Yes

Reviewer #2: No

5. Review Comments to the Author

Reviewer #1: The authors have fulfilled the requirements for the manuscript, presented the research data, and the conclusions are substantiated.

The question - if there is an effect of fertilizers on chlorophyll fluorescence, then why is there no significant effect on the amount of chlorophyll? Then what is the reason for the change in fluorescence, it only with morphological changes in the leaf?

Reviewer #2: This was very well thought out experiment to test the effects of different inoculants on early spinach growth.

I had some issues reading and understanding parts of the manuscript. An additional edit for clarity would make it easier for readers to understand the work done and discussion presented. For example in the first paragraph of the discussion lines 246-250 “In this study, the inoculations

of biofertilizer effectively improved the plant height, leaf length, and leaf weight of

spinach, among which the growth status of plants with the inoculations of Bs and Bm

was more dominant while which was similar to the results of Samia et al. [34] while

the inoculation of Ba didn’t have more effects.”. Breaking long sentences like this up would improve how well this information is presented to the readers. Similarly changing awkward wording like “while which” and “didn’t have more effects” would provide a clearer point to the discussion. Improving sections like this throughout the manuscript would greatly improve the manuscript.

To also improve clarity, I would recommend that a more consistent representation of the treatments be decided upon. In parts of the manuscript the treatments are referred to as Bs/T1 or Bm/T2. For instance, in the introduction lines 129-131 identify the treatments as multiple identifiers (Bs, T1 or Bm T2), in the conclusion on line 328 the individual inoculants are identified (Bs and Bm), Fig 1 G has the graphs as (B,C,D,E, and F), , Table 2 lists the treatments as CK,T1,T2,T3,and T4. Deciding on a consistent representation for the treatments for the figures, tables, and text will make it easier for readers to better follow the effects of the treatments.

In addition to the editing, I would recommend additional discussion of the impacts that the inoculants had on plant nitrogen status. It was mentioned in the introduction on lines 81-82 the impact that Bs has on changing the availability of nitrogen and phosphorus in the soil. Nitrogen was never mentioned again in the manuscript. Nitrogen is a major plant nutrient. Despite nitrogen status not being directly measured total chlorophyll could be used as a surrogate for plant nitrogen status. The impact the inoculants had on nitrogen would be a much better claim than the impact increased phosphorous had on ATP and NADPH production as was made in lines 264-268. With the data presented in table 3 the relationships between the various inoculants become much more apparent when look at the chlorophyll a+b data. When Bs and Bm were inoculated together (T2) the plants had more chlorophyll and likely nitrogen and phosphorous. Whereas when Bs or Bm were paired with Ba there was less chlorophyll and likely nitrogen and phosphorous. Then the impacts on growth, leaf physiological indices and chlorophyll fluorescence response can be better framed around how the inoculant was impacting the plants. Presenting the results in the discussion in this context could better emphasize the potential impacts these inoculants have on early spinach growth and the importance or emphasis of species-specific inoculant and crop pairings.

Lastly, I recommend that something be added to the manuscript that better frames the results obtained from an experiment growing spinach in a growth chamber at only 300 μmol/(m2·s). Spinach can be grown in full sunlight, 2000 μmol m-2·s-1, and discussing the impacts that the treatments in a growth chamber study with such low light on PS2 antenna could be misleading. Shade and low light treatments will inherently lead to increased PS2 antenna length. Therefore, given the experiment conducted, it would be difficult to conclude or suggest that the treatments impacted antenna length. They could have contributed to antenna length but cannot be certain of their impact.

6. PLOS authors have the option to publish the peer review history of their article (what does this mean?). If published, this will include your full peer review and any attached files.

Reviewer #1: **Yes: **Dr. Oksana Belous

Reviewer #2: No

---

## [Author Response · Author response to Decision Letter 0]

4 Sep 2023

Response to reviewers’ comments:

Re: Manuscript # PONE-D-23-13632:

Review Comments to the Author

Reviewer #1: The authors have fulfilled the requirements for the manuscript, presented the research data, and the conclusions are substantiated.

The question - if there is an effect of fertilizers on chlorophyll fluorescence, then why is there no significant effect on the amount of chlorophyll? Then what is the reason for the change in fluorescence, it only with morphological changes in the leaf?

- Accepted and Revised. In ms, the mean values were compared, and the difference in the amount of chlorophyll was also significant, but the F value was written incorrectly, we are so sorry about the mistake. When facing the external changes, the first change is the chlorophyll fluorescence, which could change the chlorophyll concentrations of the leaves and ultimately affects the morphological changes in the leaf.

Reviewer #2: This was very well thought out experiment to test the effects of different inoculants on early spinach growth.

I had some issues reading and understanding parts of the manuscript. An additional edit for clarity would make it easier for readers to understand the work done and discussion presented. For example in the first paragraph of the discussion lines 246-250 “In this study, the inoculations of biofertilizer effectively improved the plant height, leaf length, and leaf weight of spinach, among which the growth status of plants with the inoculations of Bs and Bm was more dominant while which was similar to the results of Samia et al. [34] while the inoculation of Ba didn’t have more effects.”. Breaking long sentences like this up would improve how well this information is presented to the readers. Similarly changing awkward wording like “while which” and “didn’t have more effects” would provide a clearer point to the discussion. Improving sections like this throughout the manuscript would greatly improve the manuscript.

- Accepted and Revised. We have revised in the ms.

To also improve clarity, I would recommend that a more consistent representation of the treatments be decided upon. In parts of the manuscript the treatments are referred to as Bs/T1 or Bm/T2. For instance, in the introduction lines 129-131 identify the treatments as multiple identifiers (Bs, T1 or Bm T2), in the conclusion on line 328 the individual inoculants are identified (Bs and Bm), Fig 1 G has the graphs as (B,C,D,E, and F), , Table 2 lists the treatments as CK,T1,T2,T3,and T4. Deciding on a consistent representation for the treatments for the figures, tables, and text will make it easier for readers to better follow the effects of the treatments.

- Accepted and Revised. Thank you for you comments, we have revised the representation making the ms more consistent and better understanding. In the lines 129-131, that was the material introductions. For the treatment introduce, it was on the experiment design part.

In addition to the editing, I would recommend additional discussion of the impacts that the inoculants had on plant nitrogen status. It was mentioned in the introduction on lines 81-82 the impact that Bs has on changing the availability of nitrogen and phosphorus in the soil. Nitrogen was never mentioned again in the manuscript. Nitrogen is a major plant nutrient. Despite nitrogen status not being directly measured total chlorophyll could be used as a surrogate for plant nitrogen status. The impact the inoculants had on nitrogen would be a much better claim than the impact increased phosphorous had on ATP and NADPH production as was made in lines 264-268. With the data presented in table 3 the relationships between the various inoculants become much more apparent when look at the chlorophyll a+b data. When Bs and Bm were inoculated together (T2) the plants had more chlorophyll and likely nitrogen and phosphorous. Whereas when Bs or Bm were paired with Ba there was less chlorophyll and likely nitrogen and phosphorous. Then the impacts on growth, leaf physiological indices and chlorophyll fluorescence response can be better framed around how the inoculant was impacting the plants. Presenting the results in the discussion in this context could better emphasize the potential impacts these inoculants have on early spinach growth and the importance or emphasis of species-specific inoculant and crop pairings.

- Accepted and Revised. Thank you for your comments; we have discussed more in the ms.

Lastly, I recommend that something be added to the manuscript that better frames the results obtained from an experiment growing spinach in a growth chamber at only 300 μmol/(m2•s). Spinach can be grown in full sunlight, 2000 μmol m-2•s-1, and discussing the impacts that the treatments in a growth chamber study with such low light on PS2 antenna could be misleading. Shade and low light treatments will inherently lead to increased PS2 antenna length. Therefore, given the experiment conducted, it would be difficult to conclude or suggest that the treatments impacted antenna length. They could have contributed to antenna length but cannot be certain of their impact.

- Accepted and Revised. Thank you for your comments; we have discussed more in the discussion part. For the future experiment, we will design more light treatments to analyze the impact to spinach leaves.

Overall, the manuscript in its current form is hard to understand (which took me time to read through it again and again). Thus, I strongly suggest for a language editor to go through the paper and present the data and findings in a more comprehensible yet intelligible form of standard English. The title needs to be drastically changed to a more compelling statement as it appears to be too “amateur with preliminary data”. Although when you read through the paper, it actually contains noteworthy data that would make a significant contribution to the field of agriculture. The experiments and other analysis were indeed performed to high technical standard although the statistical analysis needs to be changed from Tukey’s test to Bonferroni test which is more appropriate based on its experimental design.

- Accepted and Revised. We let a professional English speaker reviewed the paper and according to his comments we checked the entire article and revised.

---

## [Decision Letter · Decision Letter 1]

25 Sep 2023

PONE-D-23-13632R1Effects of biofertilizers on the growth, leaf physiological Indices and chlorophyll fluorescence response of spinach seedlingsPLOS ONE

Dear Dr. Zhang,

Thank you for submitting your manuscript to PLOS ONE. After careful consideration, we feel that it has merit but does not fully meet PLOS ONE’s publication criteria as it currently stands. Therefore, we invite you to submit a revised version of the manuscript that addresses the points raised during the review process.

We look forward to receiving your revised manuscript.

Kind regards,

Mayank Gururani

Academic Editor

PLOS ONE

Journal Requirements:

Reviewers' comments:

Reviewer's Responses to Questions

**Comments to the Author**

1. If the authors have adequately addressed your comments raised in a previous round of review and you feel that this manuscript is now acceptable for publication, you may indicate that here to bypass the “Comments to the Author” section, enter your conflict of interest statement in the “Confidential to Editor” section, and submit your "Accept" recommendation.

Reviewer #3: All comments have been addressed

Reviewer #4: (No Response)

Reviewer #5: All comments have been addressed

2. Is the manuscript technically sound, and do the data support the conclusions?

Reviewer #3: Yes

Reviewer #4: No

Reviewer #5: Yes

3. Has the statistical analysis been performed appropriately and rigorously? 

Reviewer #3: Yes

Reviewer #4: Yes

Reviewer #5: Yes

4. Have the authors made all data underlying the findings in their manuscript fully available?

Reviewer #3: Yes

Reviewer #4: Yes

Reviewer #5: Yes

5. Is the manuscript presented in an intelligible fashion and written in standard English?

Reviewer #3: Yes

Reviewer #4: No

Reviewer #5: (No Response)

6. Review Comments to the Author

Reviewer #3: (No Response)

Reviewer #4: The manuscript is of interest to the journal however it presents many problems related to the interpretation of the results obtained.

In the description of the results the authors never take into account the results of the statistical analysis applied to the data obtained (see comments in the text) and therefore the discussion is also misleading

In addition, some sentences do not make any sense and there is no consequential and logical discussion.

Even the interpretation of some parameters obtained with the determination of chlorophyll fluorescence reveals a lack of knowledge of the latter

Reviewer #5: (No Response)

7. PLOS authors have the option to publish the peer review history of their article (what does this mean?). If published, this will include your full peer review and any attached files.

Reviewer #3: No

Reviewer #4: No

Reviewer #5: No

---

## [Author Response · Author response to Decision Letter 1]

29 Oct 2023

Response to reviewers’ comments:

Re: Manuscript # PONE-D-23-13632R1:

Review Comments to the Author

Reviewer #4: The manuscript is of interest to the journal however it presents many problems related to the interpretation of the results obtained.

In the description of the results the authors never take into account the results of the statistical analysis applied to the data obtained (see comments in the text) and therefore the discussion is also misleading.

- Revised. Thank you for your comments; we have discussed more in the ms and some details were shown in the following pages.

In addition, some sentences do not make any sense and there is no consequential and logical discussion.

- Revised. Thank you for your comments; we have discussed more in the ms.

Even the interpretation of some parameters obtained with the determination of chlorophyll fluorescence reveals a lack of knowledge of the latter.

- Revised. Thank you for your comments; we have discussed more in the ms.

Line 60 The plant species?

- Revised. 

Line 86 ‘covert’ changed to ‘coverts’.

- Revised. 

Line 89 ‘performances’ changed to ‘indicators’.

- Revised. 

Line 98,99 ‘significant’ changed to ‘a significant’? ‘Verticillium’ should be italic.

- Revised. 

Line 127-131 when was applied?

- Revised. It was applied at March 10th, 2021.

Line 181 ‘and other growth parameters’ which parameters?

- Revised. Sorry about that, and here we didn’t analyze the other growth parameters.

Line 183 ‘was maximum under the T2 treatment’ The sentence it is not true because T2 is similar to T1.

- Revised. From the original data we can see actually it was maximum under T2 and for the statically analysis they didn’t have significant differences.

Line 185 ‘And the T2 treatment had the greatest impact on the leaf physiological responses.’ This sentence it is not true for all physiological parameters.

- Revised. From the original data we can see actually it was maximum under T2 and for the statically analysis they didn’t have significant differences.

Lime 186 ‘Compared with the F value, the difference in soluble sugar (SC) was the highest (F=118.35; P<0.01).’ this sentence has not sense.

- Revised. 

Line 192 Table 3 as it is possible that 0.58 is ab and 0.6 is b for Chl b.

- Revised. Sorry about that, here we wrote the wrong letter.

Line 201 ‘value of the T2 treatment was the highest’ this sentence it is not true.

- Revised. The F0 value under T2 was 10013, it was the highest. Sorry that we labelled the wrong letter in Table 4 and revised.

Line 205 ‘of the Fv/F0 value, the T2 treatment remained the largest’ this sentence it is not true.

- Revised. The Fv/F0 value under T2 was 4.08, which was the highest. We labelled the wrong letter in Table 4 and revised..

Line 206 ‘Compared with the F value, the difference of F0 was the largest (F=112.57; P < 0.01).’ this sentence has not sense.

- Revised. 

Line 217-218 ‘treatment had the greatest effect on the specific activity of each reaction center (RC), and there was a significant difference between the TR0/RC and DI0/RC.’ this sentence it is not true.

- Revised. 

Line 220-222 ‘The quantum yield (φP0), efficiency (φE0), and ψ0 values were highest under the T2 treatment, and with the exception of ψ0, there were significant differences (Table 6) (P<0.01).’ this sentence it is not true.

- Revised. 

Line 224-225 Table 6.

- Revised. 

Line 233-234 ‘the fluorescence value of the T2 treatment was the largest, which signified that it had a greater effect on the photosynthetic chemical rate of the leaves.’ this sentence it is not true.

- Revised. 

Line 235-236 ‘As shown in Fig 1(B), the difference of relative variable fluorescence (Vt) at point J (2 ms) was the largest.’ This is not evident in Fig 1B.

- Revised. We analyzed the Vt data at point J, the ANOVA showed it had the highest differences (F=2354.87).

Line 275-276 ‘The growth status of plants with the inoculations of Bs or Bm was more dominant and it was similar to the results of Samia et al. [34].’ this sentence it is not true.

- Revised. 

Line 281-282 ‘inoculated together, the leaf physiological indices of spinach seedling (SP, SC, Chl a, Chl b, Chl a+b, Chl a/b, and WP) were significantly enhanced.’ This is not true for Chlb and Chla/b.

- We have revised the letters in Table 3. For Chlb and Chla/b, they were also significantly enhanced.

Line 284-285 ‘The application of phosphate fertilizer can promote the synthesis of chlorophyll in leaves [37].’ What means this sentence?

- Revised. At first, we want to use the fertilizer adding can improve the synthesis of chlorophyll in leaves to explain our data. 

Line 291-293 ‘Thus promoting the synthesis of ATP (Adenosine triphosphate) and NADPH (nicotinamide adenine dinucleotide phosphate) in leaves then promoting the growth.’ This sentence has no sense.

- Revised. 

Line 294-295 ‘phytohormone biosynthesis pathways; modulate ethylene levels in plants and the launch of host plants’ systemic tolerance.’ It is not possible to hypothesize because author didn’t determine the molecules. 

- Revised. We cited the relevant research results from other studies.

Line 302-303 ‘the concentration of MDA in leaves treated with biofertilizer was decreased which mean the plant was in a suitable environment.’ Why MDA amount is so high in control?

- Maybe because in control it was under a stress condition and the MDA was higher, while after biofertilizer adding it was under a suitable condition and the MDA amount was lower.

---

## [Editor Report · Decision Letter 2]

31 Oct 2023

Effects of biofertilizers on the growth, leaf physiological Indices and chlorophyll fluorescence response of spinach seedlings

PONE-D-23-13632R2

Dear Dr. Zhang,

We’re pleased to inform you that your manuscript has been judged scientifically suitable for publication and will be formally accepted for publication once it meets all outstanding technical requirements.

Kind regards,

Mayank Gururani

Academic Editor

PLOS ONE
---

## [Editor Report · Acceptance letter]

3 Nov 2023

PONE-D-23-13632R2 

Effects of biofertilizers on the growth, leaf physiological Indices and chlorophyll fluorescence response of spinach seedlings 

Dear Dr. Zhang:

I'm pleased to inform you that your manuscript has been deemed suitable for publication in PLOS ONE. Congratulations! Your manuscript is now with our production department. 

Kind regards, 

on behalf of

Dr. Mayank Gururani 

Academic Editor

PLOS ONE